# Dynamics of Biocorrosion in Copper Pipes under Actual Drinking Water Conditions

**Carlos Galarce** [1], **Diego Fischer** [1], **Beatriz Díez** [2], **Ignacio T. Vargas** [1,3] and **Gonzalo E. Pizarro** [1,*]

[1] Department of Hydraulic and Environmental Engineering, Pontificia Universidad Católica de Chile, Santiago 7820436, Chile; cegalarc@uc.cl (C.G.); dafische.uc@gmail.com (D.F.); itvargas@ing.puc.cl (I.T.V.)
[2] Department of Molecular Genetics and Microbiology, Pontificia Universidad Católica de Chile, Santiago 7820436, Chile; bdiez@bio.puc.cl
[3] Centro de Desarrollo Urbano Sustentable (CEDEUS), Santiago 7520245, Chile
[*] Correspondence: gpizarro@ing.puc.cl

**Abstract:** Deficient disinfection systems enable bacteria to form in drinking water; these can invade plumbing systems even if the pipes are composed of antibacterial materials such as copper. Severe copper corrosion by microorganisms and their subsequent release into the water system are evidenced by the blue water phenomenon. Proper monitoring and control can reduce such undesirable effects on water quality. However, a lack of data from analysis under actual conditions has limited the development of useful predictive tools and preventive strategies. In this work, an experimental aging system was connected to a drinking water network affected by the blue water phenomenon. The microbially influenced corrosion (MIC) was evaluated by studying the dynamics of the formed bacterial community and its relationship with copper corrosion and the release of copper. The results suggest that the conformation and composition of the biofilm attached to the surface influence the measured parameters. The corrosion rate was variable throughout the sampling time, with the highest value recorded after one year of aging. The composition of biofilms also changed with time; however, the genus *Pseudomonas* was ubiquitous over the sampling time. No relationship between the corrosion rate and the biofilm age was observed, thereby suggesting that MIC is a dynamic phenomenon that requires further study.

**Keywords:** copper; corrosion rate; biocorrosion; biofilm

## 1. Introduction

Water utilities invest time and money toward controlling the presence of undesirable microorganisms in drinking water systems [1]. The microbial contamination of a water distribution network begins with microbial attachment on a pipe surface [2,3], which then forms a mixed-species biofilm [4]. Biofilm formation is a crucial element for both the promotion of corrosion and the survival of microbial communities in corroded copper plumbing [5,6] because the biofilm's structure can modify interfacial metal-solution reactions, inducing changes in the ion concentrations, pH, dissolved oxygen (DO) levels, and organic and inorganic material detachment [6,7].

Copper pipes in the drinking water service are highly valued in domestic plumbing systems owing to their corrosion resistance [7,8] and antiseptic properties [9,10]. However, bacteria living in copper plumbing systems have reported [1,7,11,12]. Microorganism settlement can promote and enhance copper corrosion in pipes [2,3], causing a phenomenon known as microbiologically influenced corrosion (MIC) or biocorrosion [6,13]. Biocorrosion leads to two problems: high levels of copper in drinking water, which is a public health concern [14–17], and infrastructure failure owing to localized

corrosion [7]. Moreover, owing to the development of extracellular polymeric substances (EPS) by the biofilm, an unknown amount of copper can be stored or sorbed on the biofilm´s surface [14,18], which increases the risk of contaminating the water with undesirable copper concentrations.

Consumption of water containing copper has adverse effects on human health, ranging from stomach distress to the liver or kidney failure [19–22]. The World Health Organization (WHO) limits the copper concentration in drinking water to 2 mg/L [23]. Nevertheless, extreme copper corrosion cases can lead to copper concentrations of up to 20 mg/L [24] manifested as blue coloring, which is referred to as the blue water phenomenon.

The difficulty in accessing the internal surface of copper pipes within operational networks creates challenges in the study of biofilms [1]. Most of the available information on biofilms in copper pipes have been collected by using sampling points or by assessing a few selected microorganisms under controlled laboratory conditions. However, neither of these methods represents the dynamics of diverse communities within real networks [5,7,25,26]. *Pseudomonas* is one of the most common types of bacteria found in the biofilm that develops in plumbing systems [2,3,27]. It has been reported that its adhesion capacity promotes biofilm development [27,28], and the secretion of enzymes with oxidative activity as a catalase is associated with the corrosive process [29]. Although these studies conducted under laboratory conditions have provided valuable information, the conditions often differ significantly from those occurring in premise plumbing, particularly in the biofilm growth cycle and temporal variability [3].

In this study, an experimental copper pipe aging system is connected to a drinking water network affected by MIC. The main objective is to investigate the bacterial community changes and the association of MIC with the dissolved copper in water. Chemical, hydrodynamic, and microbiological evaluations were conducted during a two-year experiment to address the dynamic nature of the process and to provide a comprehensive study of copper MIC under actual conditions.

## 2. Materials and Methods

### 2.1. Sample Collection

An aging system consisting of six parallel copper pipes every 1 m in length with an internal diameter of 1.95 cm (3/4 inches) and a volume of 300 mL [30] was installed in the yard of a residence in Olmué, a rural community in Chile (32°59′43″ S, 71°11′08″ W; Figure 1A), that was affected by the blue water phenomenon (Figure 1B). In Figure 1B, it is possible to appreciate the change in the watercolor (with a green-blue tone), which is the result of corrosion. The water was supplied from a well located near the residence without a disinfection system. The experimental aging system was connected to the plumbing system (Figure 1C) and was flushed three times a day (every 8 h) for 5 min. Two of the copper pipes were collected and replaced for each sample time of 1, 3, 8, 12, and 24 months. The exposure was initiated starting from August 2012 (winter season) to April 2015 (fall season). The water samples were collected, stored at 4 °C, and rapidly transported to the laboratory for analytical and experimental procedures. At each collection time, the water quality parameters were measured to analyze the stability of the system.

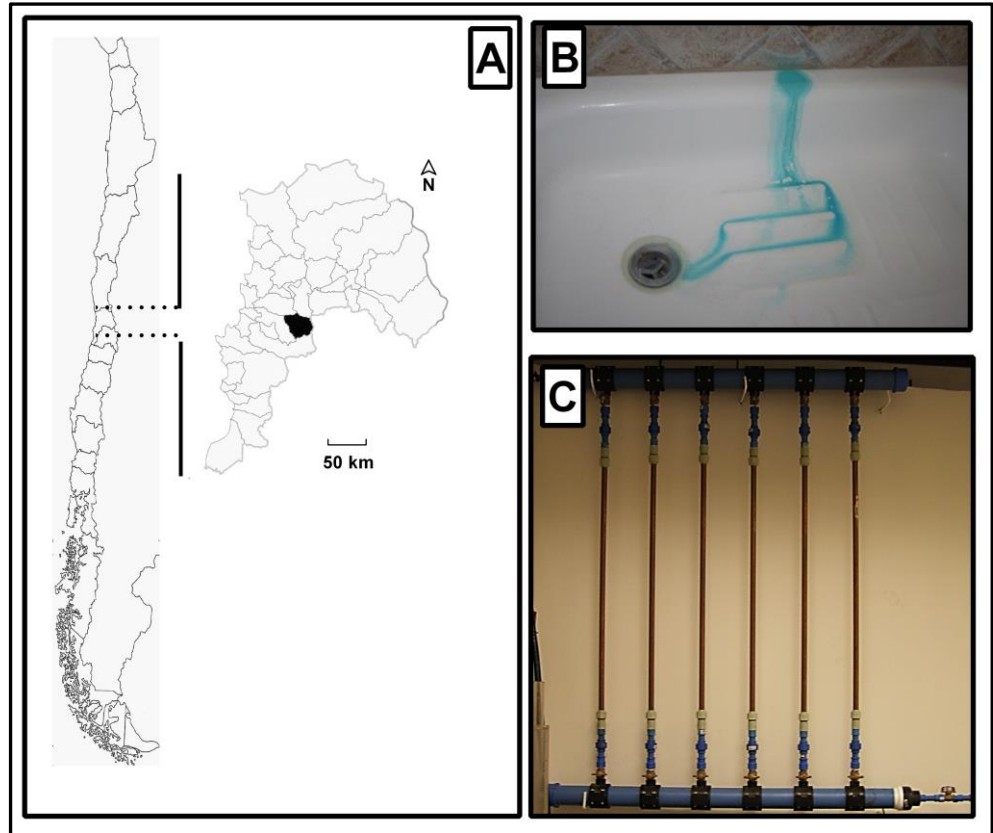

**Figure 1.** (**A**) Map of Chile highlighting the location of the sampling point in the central region. (**B**) Blue-water phenomenon observed in local premise plumbing. (**C**) Sampling system installed in the local premise plumbing system.

### 2.2. Flushing Experiments

The amount of copper released was studied using the system described in Vargas et al. [18], which consisted of a tank filled with water from the field site connected to a 1 m polyvinyl chloride (PVC) pipe followed by the sampled copper pipe. The pipe was flushed at a constant laminar flow rate of 0.48 L/min until a volume of 5 L liters passed through the system. During flushing, 12 sequential samples of 100 mL were taken at 100, 300, 500, 700, 1000, 1300, 1600, 1900, 2200, 2500, 3500, and 4500 mL. The collected samples were filtered using a 0.45 μm pore size membrane and all samples were acidified with nitric acid for preservation. The samples were analyzed by an inductively coupled plasma-optical emission spectrometer (ICP-OES; model 7300, PerkinElmer, Waltham, MA, USA) to measure the copper concentration. Two pipes were used for each sampling time.

### 2.3. Water Analysis

The water quality parameters were measured at each sample collection. Other parameters, such as pH, temperature, conductivity, and DO were measured in situ. Chemical analysis was conducted in the laboratory. Anions such as chloride, fluoride, sulfate, nitrate, nitrite, and phosphate were analyzed in the laboratory by using an ion chromatograph (IC; Metrohm, model 882 Compact IC plus, Herisau, Switzerland). The mobile phase was 3.2 mM of sodium carbonate and 1 mM of sodium bicarbonate at a flow rate of 0.7 mL/min. A Metrosep 4/5 Guard precolumn was used to support the employed Metrosep A Supp 5-250/4.0 column. The total organic carbon (TOC) was measured using Shimadzu (TOC-L CPH analyzer equipped with an autosampler (ASI-L) and a nitrogen module (TNM-L). The concentrations of copper and iron were measured using ICP-OES (model 7300, PerkinElmer, Waltham, MA, USA).

## 2.4. DNA Extraction

The copper pipes were aseptically cut into segments 25 cm in length. The inner surface of each pipe was carefully washed and filled with Milli-Q autoclaved water to avoid biofilm loss. The biofilms were released by sonication in a bath of ice water at a frequency of 40 kHz (Branson 2510 Ultrasonic Cleaner, Danbury, CT, USA). After 5 min, the sonication was paused, and the pipes were manually agitated for 10 s and then sonicated for an additional 5 min. The biofilm extracts were pelleted in 50 mL sterile polypropylene tubes by centrifugation at 12,452 *g* for 10 min. Supernatants were then withdrawn, leaving 2 mL for pellet resuspension. The pellet suspension method was employed for DNA extraction using the Power Soil® DNA isolation kit (MoBio Laboratories, Inc., Carlsbad, CA, USA) following the manufacturer's instructions [31]. Gel-Red stain was used in agarose electrophoresis to verify the obtained DNA integrity.

## 2.5. Denaturing Gradient Gel Electrophoresis

About 10 ng of extracted DNA was used as the template for polymerase chain reaction (PCR), in which 16S ribosomal DNA (rDNA) specific primers were used. For denaturing gradient gel electrophoresis (DGGE) analysis, we used primers 358f-GC and 907r [32] with a 40 bp CG clamp attached to the 5' end of the forward primer required for DGGE methodology, which amplified the fragments to a size of approximately 550 bp. The 25 μL PCR mixtures contained each deoxynucleoside triphosphate at a concentration of 100 mM, 1.5 mM $MgCl_2$; each primer at a concentration of 0.3 mM, 2.5 U of Taq DNA polymerase (New England Biolabs, Ipswich, MA, USA); and the PCR buffer supplied with the enzyme [33]. The PCR program specified an initial denaturation at 94 °C for 5 min followed by 30 cycles of denaturation at 94 °C for 1 min, annealing at 55 °C for 1 min, and extension at 72 °C for 1 min. During the last cycle of the program, the length of the extension step was increased to 10 min [33]. An aliquot of the PCR product was electrophoresed on a 1.0% agarose gel and stained with Gel-Red, and the concentration was estimated by using the Low DNA Mass Ladder as a standard (Gibco BRL, Thermo Fisher Scientific, Waltham, MA, USA). DGGE gel containing 0.6% polyacrylamide was produced at 0.75 mm in thickness with urea-formamide differential concentrations of 40%–70% as DNA denaturant agents [33]. The DGGE was run at 100 V per 16 h using SYBR Gold Stain (Molecular Probes, Thermo Fisher Scientific, Waltham, MA, USA) at 0.01% for 30 min, and the results were revealed using an ultraviolet (UV) transilluminator (Bio-Rad Technologies, Berkeley, CA, USA).

## 2.6. Sequence Analysis

To identify the bacterial community, the profile bands with the highest intensity were excised, and their DNA was amplified using the previously described primers without the 40 bp GC clamp [33]. Then, the PCR products were confirmed by electrophoresis and were sent for sequencing (Macrogen Inc., Seoul, Korea). The recovered sequences were then aligned and compared with the database of the National Center for Biotechnology Information (NCBI) using the basic local alignment search tool (BLAST; NCBI) algorithm. The 16S rRNA phylotypes retrieved from the DGGE band sequences together with the reference taxa, and the closest relatives from GenBank, included only in published studies or cultures, were aligned using MAFFT version 7.123b software. The retrieved sequences from the present study were analyzed against the phylogenetic tree containing all sequences in the Silva database (http://www.arb-silva.de/).

Phylogenetic reconstruction using maximum-likelihood search strategy with 10,000 bootstrap replicates was performed subsequently for each gene dataset using the FastTree version 2.1.9 SSE3 software.

## 2.7. Scanning Electron Microscopy

Eight coupons of 1 cm × 1 cm from were cut aseptically from the field copper pipes for microscopic analysis. The samples were kept hydrated with the corresponding test water before preparation for

scanning electron microscopy (SEM). The coupons were fixed with 2.5% (w/v) glutaraldehyde buffered with 0.1 M phosphate (pH 7). The samples were rinsed with sterile distilled water, postfixed with 1% (w/v) osmium tetroxide for 1 h, and dehydrated in 50%–100% serial ethanol and 100% acetone baths, as described by Pavissich et al. [5]. After dehydration, the coupons were dried to a critical point and were coated with gold. The morphology and structures formed on the inner surfaces of the copper pipes were analyzed by using an SEM instrument (LEO 1420VP, *LEO* Electron Microscopy, Ltd., Cambridge, UK) with an energy dispersive spectroscopy (EDS) system including an Oxford 7424 solid-state detector for elemental analysis.

### 2.8. Electrochemical Test

A three-electrode cell was used for the electrochemical tests. Several coupons of 1 cm × 1 cm were cut randomly from the field pipes for use as working electrodes. The counter electrode was composed of carbon, and the reference electrode was composed of silver/silver chloride. Electrochemical measurements were completed with a potentiostat (CHI Instruments 750D, Austin, TX, USA). Artificial tap water, similar to that reported by Feng et al. [34] (Table 1), was used as an electrolyte to compare the electrochemical responses.

**Table 1.** Chemical composition of simulated tap water used as an electrolyte in the electrochemical tests.

| Parameter | Simulated Tap Water Concentration |
|---|---|
| Chloride ($Cl^-$) | 11.4 mg/L |
| Sulfate ($SO_4^{-2}$) | 90 mg/L |
| $HCO_3^-$ | 98 mg/L |
| pH | 7.5 |
| Conductivity | 680 μS/cm |
| Temperature | 23 °C |

Linear polarization (LP) was used to evaluate the corrosion rate at each sample time, whereas electrical impedance spectroscopy (EIS) was used to check the LP values and to support the proposed model of the superficial corrosion process [34]. EIS measurements were conducted at the open circuit potential over a frequency range of $10 - 5 \times 10^{-6}$ kHz using an alternating current (AC) amplitude of ±10 mV.

## 3. Results and Discussion

### 3.1. Water Quality

The operational parameters have a strong influence on the MIC process [35–38]. Accordingly, the water source was monitored throughout the sampling. The values of the monitored parameters (corresponding to the beginning of the experiment) are summarized in Table 2. The pH, conductivity, and DO were stable with values of about 6.9 ± 0.06, 690 ± 40.8 μS/cm, and 8.3 ± 0.4 mg /L, respectively. Other parameters associated with microbial growth showed changes during the experiment. The TOC levels showed a constant increase from 0.35 to 3.04 mg/L) at the source. The same increase trend was observed for the total alkalinity, from 51 to 219 mg/L as $CaCO_3$, which increased to reach a plateau at the end of the study. The increase in total alkalinity and TOC over time can indicate possible contamination of the source water of the system by microbial growth [39]. This possibility is supported by the values of the sulfate, chloride, and nitrate levels, which were lower at the second sampling than those recorded at other times. A discrete and quick bacterial bloom likely occurred to trigger such changes. However, the microbial contamination in the source water of the system did not pose a problem in observing the microbial changes occurring on the inner wall of copper pipes. The changes in the bulk water communities do not exert an evident influence on the composition of the attached community, as discussed by Douterelo et al. [2]. Another approach considered a dilution effect owing

to a previous rain event; however, this effect was discarded because it did not feature the same change trend in all measured parameters.

**Table 2.** Physicochemical properties of the source water.

| Exposure Time (Month)/Parameter (Unit) | 1 | 3 | 8 | 12 | 24 |
|---|---|---|---|---|---|
| Fluoride (mg/L) | 0.019 | 0.035 | 0.152 | 0.035 | $-^1$ |
| Chloride (mg/L) | 21.53 | 21.45 | 23.94 | 3.23 | 19.43 |
| Alkalinity (mg/L as $CaCO_3$) | 200 | 193 | 219 | 144.6 | 51.47 |
| Total hardness (mg/L as $CaCO_3$) | 145.7 | 142.29 | 139.2 | 150.2 | 147.74 |
| Sulfate (mg/L) | 155.2 | 155 | 163.6 | 34 | 149.46 |
| Nitrite (mg/L) | 0.016 | 0.009 | 0.015 | 0 | $-^1$ |
| Nitrate (mg/L) | 10.22 | 11.1 | 9.44 | 1.32 | 10.07 |
| Phosphate (mg/L) | 0.71 | 0.3 | 0.36 | 0.17 | 0.41 |
| TOC (mg/L) | 2.1 | 1.67 | 3.04 | 0.57 | 0.35 |
| DO (mg/L) | 8.1 | 8 | 8.2 | 8.8 | 8.86 |
| pH | 6.84 | 7 | 6.96 | 7 | 6.93 |
| Conductivity | 704 | 673 | 708 | 736 | 629 |

[1] Data not measured.

The temperature of the sampling place varied widely from 2 to 30 °C owing to seasonal changes. Previous reports indicate the temperature as a relevant factor influencing biofilm development. Ling et al. [40] observed differences between the biofilm communities from household water and the tap water samples owing to seasonal changes in temperature. Similarly, Qian et al. [41] showed that temperature cycles affect biofilm formation such that higher temperature keeps the growth of nonadapted mesophilic bacteria under control [41]. In our research, the temperature likely factored in the growth of the microbial community at the source of water, as reported by Ling et al. [40]. However, the inner surfaces of the copper pipes resulted in selective pressure in the microbial composition because the microorganisms were able to attach to the metal surface, which hastened the biofilm production.

*3.2. Scanning Electron Microscopy*

Figure 2 shows the inner surface of a copper pipe affected by MIC. The superficial layer is composed of oxides and byproducts of copper corrosion, which can facilitate the generation of a microenvironment [25]. Each SEM image exposed diverse stages of biofilm development on the internal surfaces of the copper pipes. Figure 2A,C shows low development in the mineral layer with microorganisms attached to the surface. The development of a heterogeneous biofilm likely enhanced the weak oxide layer with low protective capacity. Previous research revealed that the presence of a consortium of microorganisms with the ability for biofilm production promotes the precipitation of copper hydroxide with low protective function in comparison with the abiotic condition [42]. This situation and regular replacement of water in the copper pipes every 8 h might explain the low coverage of corrosion byproducts on the pipes exposed for 3 and 12 months.

On the contrary, a more complex surface was observed in the pipes exposed for 8 and 24 months (Figure 2B,D), in which a more homogenous corrosion byproduct layer was developed that included microorganisms, suggesting early biofilm formation. Figure 2B shows the sample exposed for eight months, in which sea urchin-shaped precipitates formed. Even though a complete characterization of these precipitates was not conducted (i.e., X-ray powder diffraction analysis), the EDS analysis of the corrosion byproducts revealed a composition of carbon, oxygen, and copper. These results, together with the observed shapes and the high alkalinity of the water averaging 162 mg/L as $CaCO_3$ (Table 2), suggest the presence of copper carbonates on the surface. It has been reported that malachite $(Cu(OH)_2CuCO_3)$ has a passivating effect on the copper surface [7,25]. The passivation characteristics of malachite would explain the increments of open circuit potential (OCP) values and the lower corrosion rate shown in the samples exposed for eight months (shown later). Similarly, diverse precipitates were observed in the sample exposed for 24 months (Figure 2D). In both cases, a low number of bacteria

were attached to the surface, particularly in the surface cracks of oxide layers. The diverse exposition areas likely promoted the copper corrosion at these sampling times. This evidence suggests that the microorganisms attached to the inner pipe surfaces and that their EPS modified the corrosion process and the release of copper owing to changes in the oxide layer that formed on the inner surface of the copper pipes during the exposure time.

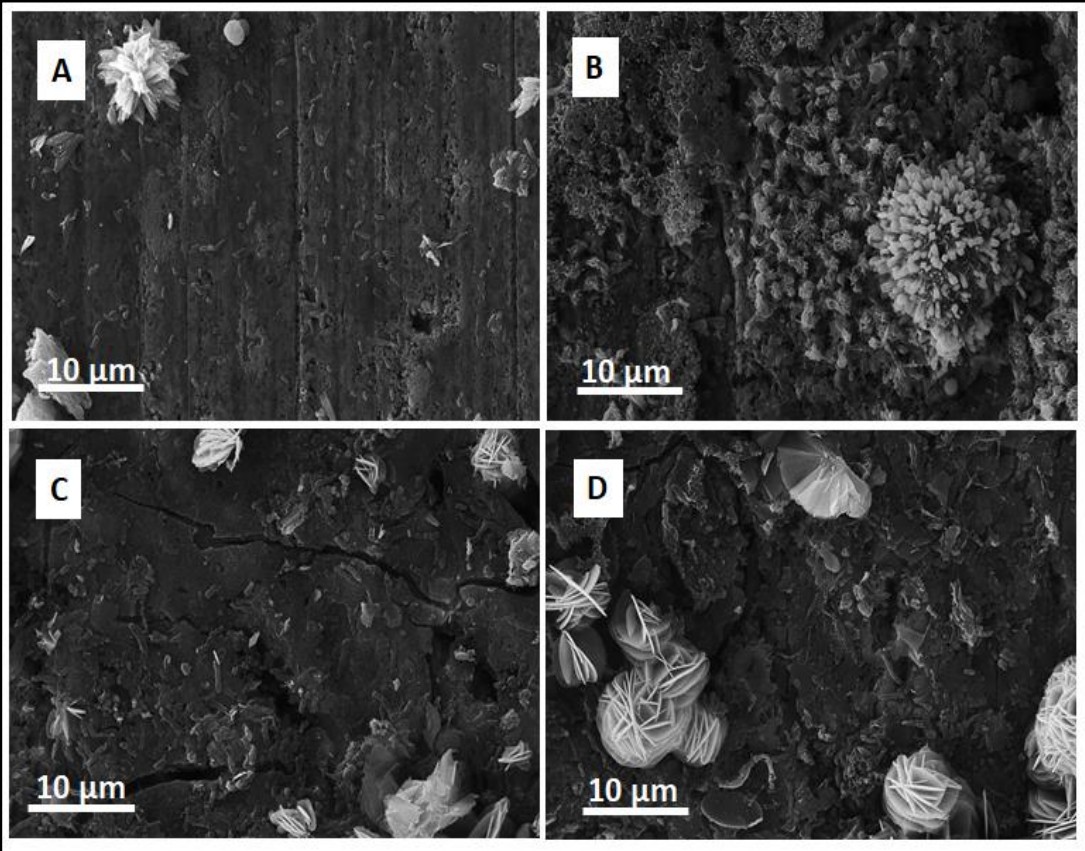

**Figure 2.** SEM images of copper after different periods of exposure: (**A**) 3 months; (**B**) 8 months; (**C**) 12 months; and (**D**) 24 months.

*3.3. Flushing Experiments*

The environmental conditions present at the sampling site were enabling to study the development of an extreme (bio) corrosion event. Moreover, the experimental set-up allowed us to observe the behavior of release copper in actual premise plumbing. The results of the flushing experiments suggest that the microorganisms attached to the inner surfaces of the copper pipes modified the copper release process under actual biotic conditions. The released copper measured in the flushing experiments could be explained in two ways: (i) by modification of the oxide layer composition and (ii) by biosorption of copper by the EPS of the biofilm. Both can be inferred after comparing our results with the data previously reported in Vargas et al. [18,25]. Modification of the oxide layer can explain the higher levels of dissolved copper in our research compared with the amount released in our previous experiments in the same range of exposure time [18,25]. The values observed in this research were more than 10 times higher than reported previously. These differences can be attributed to exposure to uncontrolled environmental conditions, which allowed us to appreciate the development of a weak oxide layer into the copper pipe that did not observe in studies before.

The weak layer was likely influenced by both the chemical composition of the water source and the microbial community attached to the copper pipe interiors. The presence of sulfate, bicarbonate, and orthophosphate enhances the formation of brochantite and cupric phosphate during long periods,

which would prevent the formation of insoluble tenorite or malachite phases [43] from creating a weak oxide layer. Moreover, previous research indicates that the presence of a biofilm creates an acidic microenvironment that hastens the formation of soluble copper compounds as cupric hydroxide $(Cu(OH)_{2(s)})$ [7,25,42,44].

The EPS influence on the biosorption of dissolved copper can be deduced from the data shown in Figure 3. After 1 L of water was extracted from the pipe, the total mass of the dissolved copper released reached a plateau for all-time series. Vargas et al. [25] reported a constant release of copper for both abiotic and biotic conditions; however, our results did not show the same behavior. A large number of EPS might have been attached to the copper surface owing to biofilm growth at the same time, which would have enhanced the release of copper corrosion products, as reported previously [25]. The results indicate that the sorption capacity of the microorganisms attached to the inner surface of the copper pipes controls the release of copper at the end of the flushing experiment. Similar results were not observed in previous studies likely because the testing conditions did not provide consecutive sampling points for reviewing the microbial succession and its effects on copper corrosion. Moreover, the field conditions might have supplied the requirements or signals needed to change the EPS production, which would have modified the corrosion response.

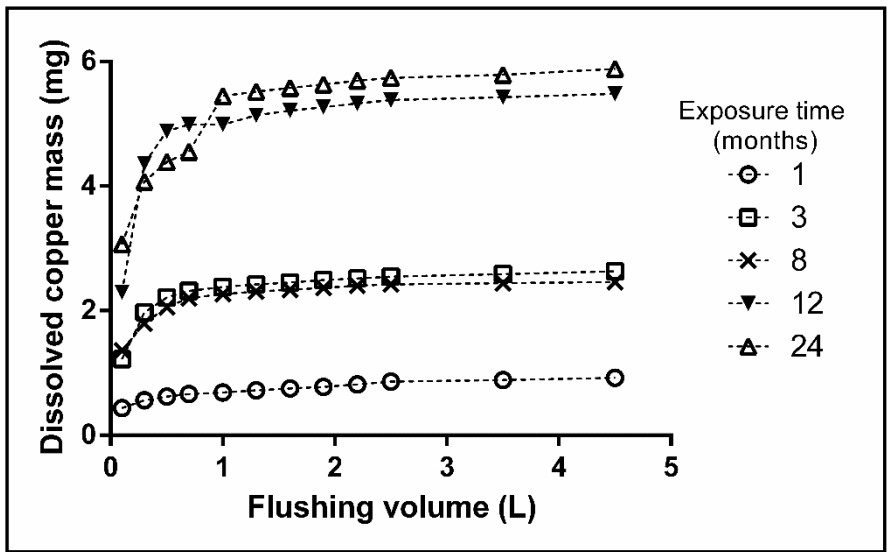

**Figure 3.** The average value of dissolved copper mass released from the copper pipe during flushing experiments.

On the contrary, Feng et al. [45] showed a direct relationship between the immersion time and the copper concentration during 21 days in neutral tap water. However, the analysis of dissolved copper did not show the same relationship in our research. The flushing experiments revealed three levels of dissolved copper in three discontinuous ranges of time after 1 L of water was passed through the pipe. The first level was reached in the pipe exposed for one month, with an average value of 0.5 mg of dissolved copper after 8 h of stagnation. The second level was achieved in the samples exposed for three and eight months, which had similar copper levels of about 2.2 mg and exceeded the WHO recommendation for the first liter of water flushed in the pipe [46]. Finally, an increment of dissolved copper was observed in the samples exposed for 12 and 24 months, both of which had an accumulated mass release of about 5 mg. Probably the shifts on copper release between the samples exposed to 8 and 12 months are the result of a different stage of biofilm development for each time. Additionally, an active replacement of biofilm (attachment/detachment) exposed the metal to the bulk water again, which might increase the release of copper after 8 months. The exposure under actual environmental conditions, allowed us to observe the different levels of copper released that was not appreciated in previous reports.

Consistently, after three months, the amount of dissolved copper mass and the peak concentration after passing 1 L of water through the pipe in the flushing experiments exceeded the WHO recommendation, which is considered to be harmful to humans [46]. Incorporation of the environmental and microbial aspects revealed in our research explains the differences from the results reported by Feng et al. [45]. Moreover, it possible that the outcome of Feng et al. showed the initial process of corrosion, in which the chemical kinetics control the copper corrosion. Therefore, our results present biocorrosion as a dynamic process, whereby the biofilm´s characteristics can modulate the copper release.

### 3.4. Microbial Community Analyses

Analysis of the development of the biofilm attached to the inner copper pipe was conducted by DGGE fingerprinting (Figure 4). This approach revealed the manner in which microbial populations change with each tested condition. Thus, DGGE band profiles were assigned to the different phylotypes present on the DGGE gels, which enabled screening of the enriched microbial populations under the tested conditions. Knowledge of the type of bacteria attached to the surface for each sampling time will help to understand the stage of biofilm development and the possible mechanisms involved in the copper corrosion. The technique used in this analysis enabled the identification of several of the most abundant/dominant members of bacteria. The limited DNA concentrations obtained for most of the samples over time prevented a more detailed analysis.

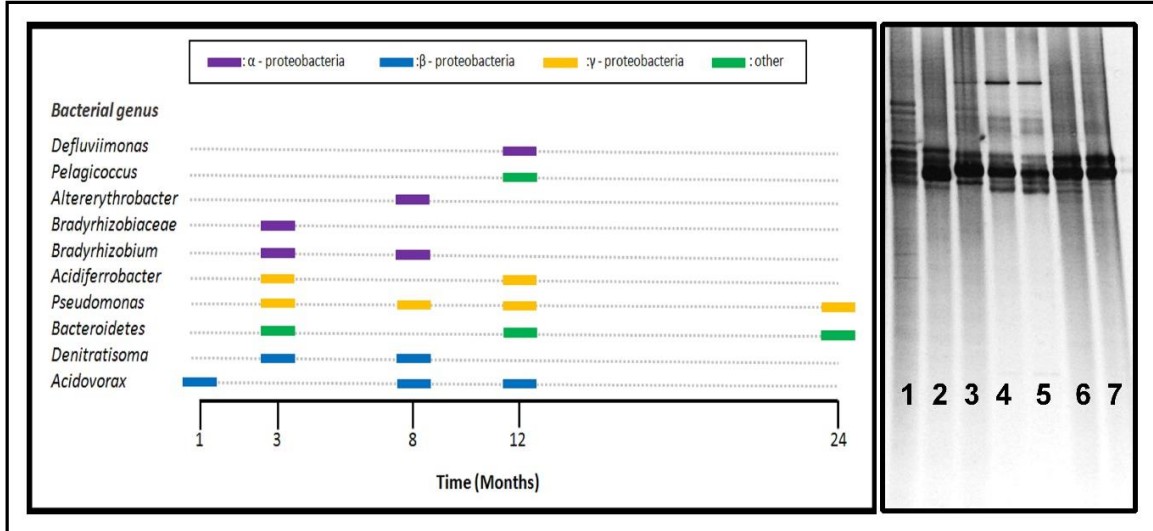

**Figure 4.** (**Left**) Diagram showing the distribution of bacterial phylotypes over time. (**Right**) Denaturing gradient gel electrophoresis (DGGE) profile of 16S rRNA of total bacterial community from bulk (line 1) and biofilm from inner copper pipes (line 2:1 month; lines 3 and 4:3 months; line 5:8 months; line 6:12 months; and line 7:24 months).

The results indicate the occurrence of active biofilm succession/dynamics during the testing time (Figures 2 and 4). In drinking water systems, changes in the microbial community occur after even a few hours of stagnation [47]; however, the populations that are part of the biofilm's core essentially do not change during the stagnation [1,48–51]. For this reason, the stagnation time was ruled out as the primary change factor. Contamination of the water source could also explain the microbiological changes in the biofilm; however, this possibility was also discarded on the basis of Douterelo et al. [2], who showed that the local biofilm community prevailed rather than the planktonic community or an external selective pressure [2]. This means that the inclusion of a new biofilm's members depends on the selection process of the microbial community (or core) that develops the biofilm. Abiotic factors such as temperature, water quality, and aging time [47,52,53] had stronger influences on the active

renewal of biofilm because they have a greater effect on the physicochemical and biological process in the environment.

The DGGE fingerprints of the biofilm communities attached to the inner surfaces of the copper pipes showed that members of the genus *Pseudomonas* (Gammaproteobacteria) were the most persistent phylotypes during the sampling time (Figure 4). Formation of a mature biofilm can depend on the presence of *Pseudomonas* because they have a strong affinity and easy adherence to surfaces, which facilitates the attachment of other microorganisms [2,27,28]. The mechanisms of *Pseudomonas* associated with copper corrosion are related to the capacity of the biofilm formation and its enzymatic activity. Regarding the latter, *Pseudomonas* has both oxidase and catalase activity, which generate a constant cycle of oxygen renewal on the surface of a copper pipe, thereby promoting an increase in the cathodic current and thus an increase in the overall corrosion process [54–56]. Our results suggest that *Pseudomonas* form the main core of biofilm owing to their prevalence and other characteristics. The control of *Pseudomonas* attachment on a metallic surface would be an appropriate strategy for reducing or controlling the copper corrosion. Other studies have reported a similar prevalence of *Pseudomonas* in drinking water systems concerning corrosion and biofilm production [2,38,57–60].

On the contrary, the metabolism of most of the other phylotypes present in these communities and recovered by DGGE are known to be involved in the nitrogen cycle; these include *Defluviimonas*, *Bradyrhizobium*, *Denitratisoma*, and some members of Bacteroidetes [61–64]. Kelly et al. [3] reported differences in bacterial abundance related to the availability of inorganic nutrients in drinking water, specifically nitrogen, and found a significant correlation between the bacterial cell numbers and the nitrate concentration [3]. The nitrification could promote metal corrosion owing to pH reduction, which increases the solubility of minerals on the surfaces of materials [65]. Moreover, some nitrate-reducing bacteria (NRB) strains have the ability to induce NRB-assisted corrosion in other materials under anoxic conditions [66]; thus, similar behavior can be expected for copper. For these reasons and from a practical perspective, specific control of the nitrogen (nitrite and nitrate) concentration in the water would be an appropriate strategy for avoiding biofilm development and copper pipe corrosion [67,68].

Other phylotypes present at various sampling times include members of the Bacteroidetes phylum. Bacteroidetes members are commonly found in drinking water systems [40,59,69] They are a very diverse heterotrophic bacterial group with aerobic and anaerobic members that are able to degrade biopolymers [70] related to degradation of the complex compounds derived from dead biomass [71]. These phylotypes could influence copper corrosion through the stimulation of a cathodic reaction by consuming electrons as the metabolic energy sources [66]. In oxic environments, a cathodic reaction represents a reduction in oxygen, although nitrate reduction is a major microbial metabolism process in anoxic environments. Bacteroidetes members likely have less of an influence on copper pipe corrosion when they occur deep in the biofilm, where the oxygen concentration is limited.

The biofilm development stages did not show a direct relationship with copper corrosion. Vargas et al. [25] presented evidence of the differences between the biofilm community structures of field pipes and pipes tested in the laboratory. The results suggest a probable ecological succession over time, which was resolved in the present work. This conclusion is important because the order in which surfaces are colonized by either aggressive or protective bacteria influence the outcome with regard to MIC [72]. Moreover, Vargas et al. [25] proposed that the age of the biofilm determines its effects on copper corrosion. However, our results indicate that the type of bacteria present on the surface is more important because the corrosion rate is strongly influenced by the kind of oxide on the surface, which depends on the specific activity of the microbial community.

To conclude the microbial analysis, a few groups of microorganisms were detected in specific sampling times (Figure 4). These bacteria, such as those of genus *Acidiferrobacter*, are able to decrease the water pH [55,73], which helps to increase both the corrosion rate and amount of copper released. However, the *Acidovorax* phylotype has shown a positive influence on copper corrosion and the formation of EPS [73]. The secretion of EPS and enzymes creates a hospitable zone for the establishment of areas with different potentials, thereby promoting an increase in the cathodic current and, thus,

an increase in the overall corrosion process [29,54]. The type of enzyme secreted is more relevant in stress events because under such conditions, the hydrogen peroxide ($H_2O_2$) concentration increases, which triggers the secretion of catalase or peroxidase. The activity of these enzymes produces an increment of local oxygen concentration, which increases the kinetics of the cathodic reaction and, thus, the global corrosion process [17].

Finally, our microbiological analysis is the first temporal local study to address the changes in microbial biofilm communities, which allows for evaluation of the temporal prevalence of the diverse phylotypes described previously in water distribution systems [1,2,52,59,74–77].

## 3.5. Electrochemical Tests

The evolution of the copper corrosion was evaluated through electrochemical techniques. Due to its simplicity, OCP has been used in numerous studies of MIC [78]. The changes in potential observed with this technique according to a plot of potential versus time enables the detection of an accelerated attack caused by bacteria [78]. In this work, the OCP measurements showed a positive trend during the pipe aging. This result means the metal surface was ennobled, which indicates the formation of a semi-protective film composed of oxides and microorganisms that can influence the reaction of oxygen reduction on the metal surface [78] (Figure 5A). The most positive OCP value was slightly higher than zero ($0.03 \pm 7 \times 10^{-4}$ V) and was reached in the sample aged for eight months, whereas the most negative value was achieved in the sample aged one month ($-0.22 \pm 0.025$ V). The pitting potential was not detected during the study because the potential did not exceed the critical value of 0.10–0.17 V, as reported by Cornwell et al. [79].

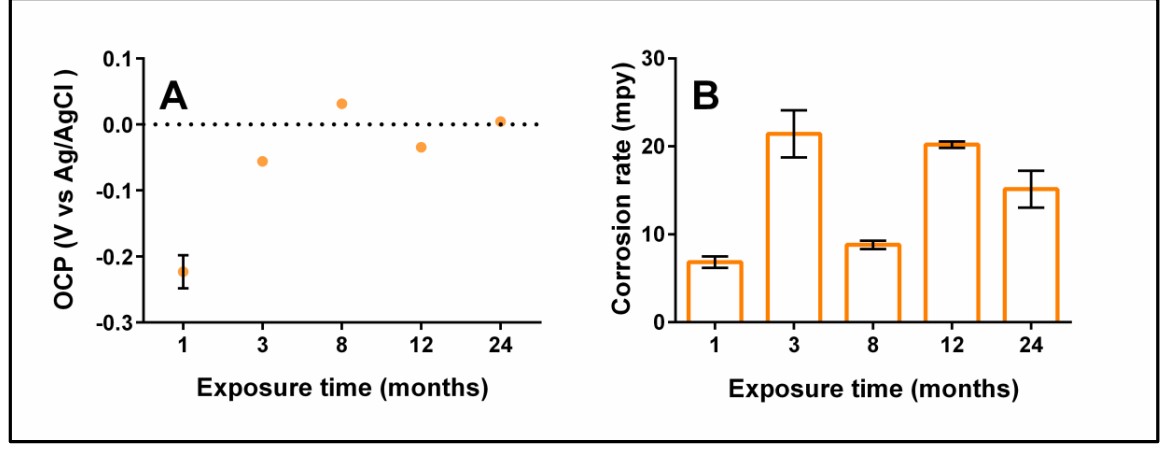

**Figure 5.** Summary of electrochemical results. OCP evolution (**A**), and variation in the corrosion rate (**B**) over time.

On the contrary, the results of linear polarization resistance (LPR) used to calculate the corrosion rate indicated a volatile value (Figure 5B). In the present study, the corrosion rate was almost seven times higher than for the unused pipes, from 3.4 to 20.2 mpy. The highest value was reached in two sampling times, but likely the mechanism was different for each one. Probably, after three months of exposure, several micro galvanic cells were formed on the pipe's surface, producing a high corrosion rate. On the other hand, in the sample aged for one year, the high value of the corrosion rate is due to the attachment/detachment process of biofilm settling down. The variability of the data could be attributed to multiple patterns in the pipe, where the microorganisms are distributed heterogeneously on the copper pipe surface.

It should be noted that the OCP values showed a trend to become stable during the exposure time, but the corrosion rate did not show the same behavior. Moreover, samples with more positive OCP values did not show a lower corrosion rate, which suggests that a reactive barrier produced by

microorganisms, EPS, and corrosion products formed during the pipe aging controlled the copper pipe corrosion.

Our results suggest that the corrosion rate was modified by the presence of microorganisms. Similar results were reported by Vargas et al. [25], in which the corrosion rate and the amount of copper released were compared under biotic and abiotic conditions. The assessment of both conditions suggests that the age of the biofilm determines its effect on copper corrosion [25]. However, the possible role of diverse biofilm members should also be considered because a small change in the biofilm members can produce a relevant change in the corrosion rate, as is described in this research.

Figure 6 shows the most representative Nyquist diagrams of copper pipes at different exposure times. The results indicate that a copper ion diffusion process controlled the corrosion, as implied by the Warburg impedance observed in the Nyquist plots. However, the diffusion process showed different behaviors at each sample time. At the initial times, the diffusional behavior observed was close to that reported by Feng et al. [34] at a pH of approximately 5 under the abiotic condition with 24 h of immersion [34], although a slight variation in the Warburg impedance angle was indicated. Frateur et al. [80] discussed the changes in Warburg impedance during the immersion time. Their results showed changes in the Warburg angle from 45 to 22.5° with increases in the immersion time under abiotic conditions [80], which was attributed to the electrode behavior as a semi-infinite porous medium. The impedance results in the present study reflect the development of oxide layers with a different type of porosity owing to the oxide layer composition of, which included a combination of biofilm and diverse corrosion products. This is supported by the angle values (Figure 6) and SEM images (Figure 2). A specific electrochemical behavior for each sampling time cannot be assigned directly because a heterogeneous surface was observed, which produces variations on the magnitude of $Z_{Re}$ and $Z_{im}$ values. However, the shape of the curve presented in Figure 6 is representative of each time.

Moreover, our results question the possibility of establishing a direct relationship between the corrosion rate and the age of biofilm using EIS measurements because the diversity of the pores on the reactive barrier and the chemical composition of the oxides could be modified by the action of a specific microorganism. The porosity modification on the reactive barrier can change the effective area of measurement and its apparent steady-state; thus, measurements by EIS could not be compared for different sampling times. The resolution of these concerns, further studies must be conducted to determine the types of oxides produced and to examine the changes in porosity of the oxide layer caused by microbial activity.

Finally, owing to the dynamic process of MIC, the inclusion of additional parameters for the physical model did not help to solve the problem by allocating an appropriated equivalent circuit because each new parameter lacked physical representation and was determined with less confidence; similar results were reported by Frateur et al. [80]. In our case, the changes in the surface could have caused short-term fluctuation of the electrochemistry at the metal–biofilm interface, which in turn might have caused a localized attack under the biofilm. Thus, biofilm can hinder the detection of potentially important smaller-scale dynamics [77,78].

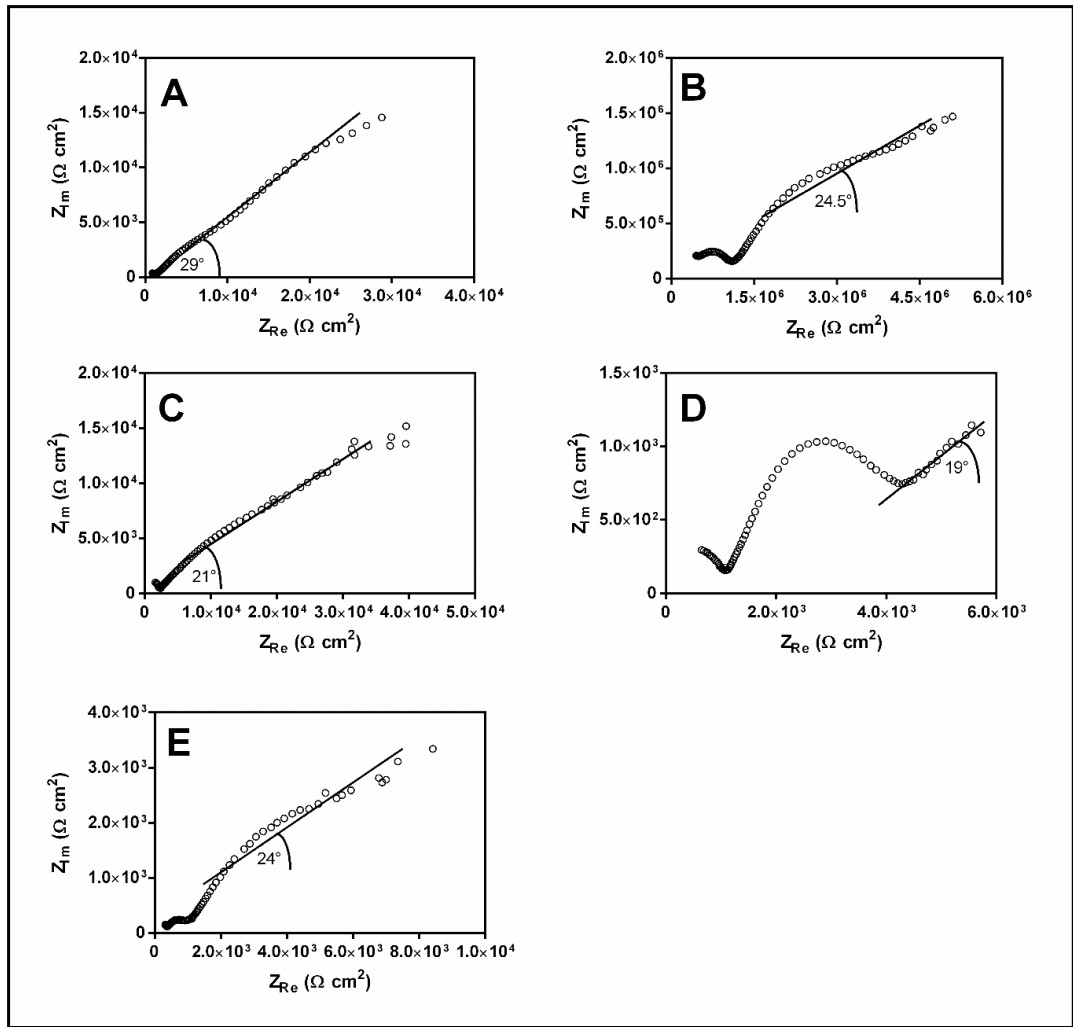

**Figure 6.** Summary of more representative electrochemical impedance results of samples exposed to (**A**) 1 month, (**B**) 3 months, (**C**) 8 months, (**D**) 12 months, and (**E**) 24 months of aging.

## 4. Conclusions

This work highlighted the relevance of studying MIC as a dynamic process. The results did not show a constant relationship among the exposure time, corrosion rate, and amount of copper released into the water. This finding revealed that the variability of a dynamic biological process could not be characterized by a single aging time or by extrapolation of short-term experiments, particularly in an actual plumbing system. Indeed, our results pose new questions and challenges in understanding the selection process that defines biofilm composition, structure, and corrosivity. Further research is required to understand the specific succession/interaction mechanisms among members of the biofilm. This could be accomplished by using advanced sequencing techniques or by considering appropriate time scales and environmental parameter monitoring over time. Finally, a better understanding of MIC processes will enable the development of appropriate control strategies, which will prevent plumbing deterioration and reduce the risk of drinking water contamination by released copper.

**Author Contributions:** Experiments were performed by C.G. and D.F. Data analyses were conducted by all authors. The experimental setup was discussed by C.G., D.F., B.D., I.T.V., and G.E.P., C.G., D.F., B.D., I.T.V. and G.P. contributed to the preparation and review of the manuscript. All authors have read and agreed to the published version of the manuscript.

**Funding:** This research was funded by CONICYT-PCHA/Doctorado Nacional/2013-21130365, FONDECYT Grant 1181326/2018, FONDECYT Grant 1150171, and the CEDEUS center CONICYT/FONDAP/15110020.

**Acknowledgments:** Special thanks are extended to Blanca Aguila Llanquilef and María Estrella Arias for providing laboratory assistance and sample preparation for DGGE assay. In addition, the authors thank Gustavo Jeria for facilitating the installation of the sampling system at the testing location.

**Conflicts of Interest:** The authors declare no conflict of interest.

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
