# Peer review of "Dynamics of Biocorrosion in Copper Pipes under Actual Drinking Water Conditions"

_water, doi:10.3390/w12041036_

Round 1
Reviewer 1 Report
This manuscript describes investigations into relationship between copper corrosion and biochemical materials in residence water pipes. In the introduction, the authors mentioned the background of this study clearly. In order to assess safety level of drinking water in residence scientifically, new methods to quantify copper corrosion and biochemical parameters should be established within real water networks. The experimental methods are also fine. The authors measured several chemicals and microbiological parameters by classical methods, which are comparable with previous studies.
A problem is the experimental results. First, I should remark a crucial mistake in Table 2. The order of the date and the exposure time are not consistent with the main text.
But, unfortunately, even if the order of the date and the exposure time is modified, I could not agree with the results and the conclusion.
I would like to point out followings.
-- The data was acquired more than 5 years ago. Nevertheless, only one replicate on each date is shown in this manuscript. Apparently, the experiment on each exposure time started from a different time point and condition, and the seasons when the data was taken were also different.
-- As written in the main text, the results shown in figures (Fig. 2, 3, 5, 6) vary and are not continuous in time series. These seem too unreliable to conclude the variability of dynamic biological process in copper tube, because we are not sure whether the variability came from biological randomness or dispersion of experimental conditions.
-- I understand the new point of this study is use of real residence water pipe. But the difference of the pipe between in laboratory and in real residence is not mentioned.
My suggestion are
-- take continuous experiments with more replicates to show the variability of this results.
-- modify the text at p. 11, l. 404-412 which includes contradiction to Fig. 5.
-- denote concretely which conditions in these experiments are different from previous researches.
This manuscript requires major revision. The authors suggested reasonable research aim and method. However, the results and conclusion are not acceptable.
Table 2 includes a crucial mistake of the order of experimental date and exposure time, which can make this manuscript completely worthless. I hope it is just a simple mistake. But, even if the order of date and exposure time are fixed, I cannot agree with the conclusion.
The authors claim their data showed variability of relationship between copper corrosion and biofilm age. However, their results cannot prove whether the variability came from the true biological randomness or the roughness of their experiments (few replicates, different season time points). I think this is a scientific defect.
When the authors mention citations, they just say ‘similar’ or ‘different’ without concrete similar or different points. This is important to clear novelty of this study as a scientific journal paper.
Reviewer 2 Report
Line 54, 55:
“The difficulty in accessing the internal surface of copper pipes within operational networks makes
creates challenges in the study of biofilms”. It seems one of the verbs should be deleted either “makes” or “creates”.
For the first figure, is it possible to show the sampling system used in the study instead of the CAD model? Also, figure 1B is not illustrative, needs more explanations on the figure representing what that blue color is, etc.
The authors specified the exact location of the used pipe. How much does the location of this study be important to achieve the final results? How the results can get globalized?
In figure 3, I suggest changing the names of the axis to “Extracted Volume” and also “released copper”.
The authors should explain why between 3 and 8 months (duration of 5 months), the value of released copper does not change that much while between 8 and 12 months (Duration of 4 months”, it is raised so much.
Please increase the resolution of figure 4.
Figure 5 is vague, as the signs were not defined. Please present it in a different mode, or with more explanations.
For figure 6, would you please check the values in the y-axis are correct? These results show that Zm is increasing from the 3rd to the 8th month, then decreasing in the 12th, and increasing once again from 12th to the 24th month! This means the behavior of the pipe is improving after a while in terms of electrochemical impedance which in reality it is not correct! Also, the temperature values are not consistent, changed from one month to another, so the authors should explain this change as well. Please add celsius after the degree sign.
Round 2
Reviewer 1 Report
I have read your response comments. I accept this version.
Reviewer 2 Report
No comment. Thanks.